# The Effect of Socio-Demographic Factors in Health-Seeking Behaviors among Bangladeshi Residents during the First Wave of COVID-19

**DOI:** 10.3390/healthcare10030483

**Published:** 2022-03-04

**Authors:** Taslin Jahan Mou, Khandaker Anika Afroz, Md. Ahsanul Haq, Dilshad Jahan, Rahnuma Ahmad, Tariqul Islam, Kona Chowdhury, Santosh Kumar, Mohammed Irfan, Md. Saiful Islam, Md. Fokhrul Islam, Nihad Adnan, Mainul Haque

**Affiliations:** 1Department of Microbiology, Jahangirnagar University, Savar, Dhaka 1342, Bangladesh; moumicro@juniv.edu; 2Monitoring, Learning and Evaluation, CEP, BRAC, Dhaka 1212, Bangladesh; anikaafroz@gmail.com; 3Gonoshasthaya-RNA Molecular Diagnostic & Research Center, Dhanmondi, Dhaka 1205, Bangladesh; shohag.stat02@gmail.com; 4Department of Hematology, Asgar Ali Hospital, 111/1/A Distillery Road, Gandaria Beside Dhupkhola, Dhaka 1204, Bangladesh; dilshad@asgaralihospital.com; 5Department of Physiology, Medical College for Women and Hospital, Dhaka 1230, Bangladesh; rahnuma.ahmad@gmail.com; 6Department of Community Medicine, Gonoshasthaya Samaj Vittik Medical College, Savar, Dhaka 1344, Bangladesh; drtariqul@gmail.com; 7Department of Pediatrics, Gonoshasthaya Samaj Vittik Medical College and Hospital, Dhaka 1344, Bangladesh; konaonu96@gmail.com; 8Department of Periodontology and Implantology, Karnavati School of Dentistry, Karnavati University, Adalaj-Uvarsad Rd, Gandhinagar 382422, Gujarat, India; santosh@ksd.ac.in; 9Department of Forensics, Federal University of Pelotas, Pelotas 96020-010, RS, Brazil; irfan_dentart@yahoo.com; 10Department of Public Health and Informatics, Jahangirnagar University, Savar, Dhaka 1342, Bangladesh; islam.msaiful@outlook.com; 11Institute of Quantitative Biology, Biochemistry and Biotechnology, School of Biological Sciences, University of Edinburgh, Murchison House, 10 Max Born Crescent, Edinburgh EH9 3BF, UK; m.f.islam@sms.ed.ac.uk; 12The Unit of Pharmacology, Faculty of Medicine and Defence Health, Universiti Pertahanan, Nasional Malaysia (National Defence University of Malaysia), Kem Perdana Sungai Besi, Kuala Lumpur 57000, Malaysia

**Keywords:** health-seeking behavior, COVID-19, socio-demographic factors, awareness, health-disparity, low-and-middle-income country

## Abstract

Background: Socio-demographic variability among nations and cultures highly influences health-seeking behavior (HSB) in managing endemic or pandemic diseases. The present study aimed to investigate the influence of socio-demographic factors on HSB among Bangladeshi residents during the first wave of COVID-19. Methods: A cross-sectional online survey was performed with Bangladeshi residents (60% male and 40% female; age range: 10–60 years or above) from May to July 2020. Information was collected from the participants who resided in slum areas or did not have internet access through face-to-face surveys, maintaining spatial distancing and proper preventive measures. A self-reported and structured questionnaire, including socio-demographic and HSB, was undertaken, and the data was analyzed using a convenience sampling method. Finally, among the 947 participants, 20 were selected using a stratified random sampling technique for in-depth-interview (IDI). The linear regression analysis was performed to determine the influence of socio-demographic factors on HSB. Results: Only about 4.2% of respondents did not wear masks, but nearly half of the participants (46.6%) did not use hand gloves. The mean score of HSB was 9.98 (SD = 2.01) out of 16, with a correct overall rate of 62.4%. As per regression analysis, higher HSBs were found among participants who reported older age (>50 years) (9.96 ± 2.45), educated unemployed students (10.1 ± 1.95), higher education (10.5 ± 1.76), and higher-income (10.4 ± 1.59); in contrast, participants living in slum areas (8.18 ± 2.34) and whose source of income was business (8.46 ± 2.04) exhibited lower HSBs. Females, compared to men, showed better HSB in every aspect, apart from online food ordering. Qualitative data showed that the younger generation is more aware because of their access to information and persuaded the older generations to follow health-seeking protocols. The results also showed that some lower-income slum-dwellers have access to information and healthcare through their employers. IDIs also found cultural, religious, and mental-health affect people’s adherence to health-seeking guidelines and regulations. Conclusions: The findings suggest that socio-demographic factors significantly influenced HSBs during COVID-19 in the Bangladeshi population. Authorities can use these observations to systematically manage future endemics or pandemics.

## 1. Introduction

The COVID-19 pandemic commenced in Wuhan, China, at the end of 2019, and since then, it has expanded to 220 countries, areas, or territories and caused a significant number of deaths worldwide [1,2]. As of 10 February 2022, 399,600,607 confirmed cases of COVID-19, including 5,757,562 fatalities, and a total of 10.09 billion vaccine doses have been administered worldwide [3]. The emergence of COVID-19 has intensified pressure on the global healthcare system and its healthcare workers (HCW), especially in resource-limited countries [4]. Bangladesh is the most densely populated developing country globally, with 21.8% of its citizens living below the poverty line [5,6]. The population density in Bangladesh is 1265 per Km^2^, and 39.4% of people live in urban areas [7,8]. The healthcare system in Bangladesh is not well prepared to provide adequate support in any emerging crisis. Unfortunately, medical facilities are more concentrated in urban areas, which deprive the people in rural areas in terms of sufficient levels of care [6,7].

In Bangladesh, the first case of COVID-19 was detected on 8 March 2020, according to the Institute of Epidemiology, Disease Control and Research (IEDCR), Dhaka, Bangladesh, and the case numbers increased steadily before the nation’s first two deaths were reported on 18 March [3]. The country had reported 1,887,271 confirmed cases of COVID-19 with 28,703 deaths on 10 February 2022 [3]. In Bangladesh, many medical and public healthcare services have been compromised due to insufficient equipment for COVID-19 patients, such as intensive care unit (ICU) beds, bedside oxygen, pulse oximeters, ventilators, and personal protective equipment. During the confinement period, private hospitals and clinics did not provide any services for fear of contact transmission [9]. As a result, healthcare facilities for primary and intensive care patients have been exhausted. The medical staff who treated the patients and were infected were criticized by society and faced social stigma from the locals [10]. There have been protests against establishing isolation facilities, hospitals, and COVID-19 care clinics in many places.

Globally, different countries have taken preventive measures to combat the pandemic [11,12,13,14,15]. Bangladesh’s government has also embraced similar safeguards, which include lockdown, social distancing, the closing of educational institutions, working from home, widespread awareness campaigns for handwashing practices, the requirement to use masks in public places, imposing regulations on international travel, establishing quarantine centers, mandatory quarantining for suspected cases of COVID-19, and the isolation of confirmed cases [16]. Vaccination is expected to be the most effective strategy in the fight against COVID-19 [17,18,19]. A nationwide vaccination program started in Bangladesh on 7 February 2021, which was free of cost and available to the people irrespective of their economic condition [3]. As of 3 February 2022, 60.37% and 38.6% of the Bangladeshi population had received their first and second doses of the COVID-19 vaccine, respectively. In contrast, only 2.82% of the population had been administered the third dose [3]. Despite all government efforts, Bangladesh had been experiencing a second wave of infection from 1 March 2021 [2,16].

People’s deficit of positive attitude toward the health safety practices regarding COVID-19 infection has made managing the pandemic challenging [20,21], especially in lower- and middle-income countries (LMICs) like Bangladesh [16,22]. Public awareness is the prerequisite to implementing the control interventions for combatting the COVID-19 pandemic; however, such awareness is governed by an individual’s health safety practices and attitudes [3,22]. It is predicted from various epidemiological models that without preventive measures to contain the spread of COVID-19, the world would confront a steep rise in the number of cases [23,24,25,26]. These scenarios have driven numerous nations to take steps aimed at “flattening the curve” to dodge a sudden surge in COVID-19 cases. The steps taken include social distancing, slowing the transmission rate of COVID-19 infections, and giving health systems more time to cope with infected individuals [27,28]. Additionally, social distancing profoundly impacts well-being in different aspects, including employment and family interactions [29,30,31,32].

A population’s health-seeking behavior (HSB) is one of the main determinants of a country’s health and socio-economic development [4,33]. HSB could be defined as “A people’s inaction, procrastination or action is undertaken following recognition by themselves of departing from good health or having a particular health problem to finding an appropriate remedy to restore health” [4]. The COVID-19 pandemic can exacerbate unhealthy medical-seeking behaviors, as delays in seeking medical care have been determined to negatively affect patient morbidity, mortality, and health outcomes [33,34,35]. People in low- and middle-income countries have been observed to have poor healthcare-seeking behaviors, thereby potentially worsening the pandemic.

Therefore, appropriate HSB is pivotal to implementing interventions and public health policies to reduce the spread of COVID-19 transmission [19,36]. HSBs of the developing nations would allow for the identifying and prioritizing resource-efficient disease management approaches to control the spread of COVID-19 in their countries [37]. Likewise, it is essential to identify the coping strategy to deal with the challenging circumstances associated with the ongoing COVID-19 pandemic or other health crises [38]. The utilization of coping strategies and HSB is influenced by different factors such as demographics, gender, pre-existing health conditions, type of employment, education level, ethnic practices, and disease patterns [39,40,41].

However, no study in Bangladesh has focused on coping strategies and HSBs among community people during the pandemic. There is a lack of information regarding triggering factors that makes the COVID-19 pandemic more dangerous to a particular community in this country. Therefore, it requires the analysis of the COVID-19 pandemic and preventive measures using well-established principles and methods that consider the complex interplay between socio-economic determinants, demographic features, and health discrepancies.

### Objectives of the Study

The present study aimed to assess the HSBs of Bangladeshi residents associated with socio-demographic factors during the first wave of the COVID-19 pandemic. The HSBs, including public adherence to COVID-19 preventive measures and lockdown protocols, will focus on finding any relevance with their demographic and sociological indicators such as age, gender, education, occupation, monthly income, residence, etc.

## 2. Materials and Methods

### 2.1. Participants and Procedures

This was a cross-sectional survey carried out involving Bangladeshi people. Mixed method research was conducted to complete the study. For quantitative data, a structured survey questionnaire with some open-ended questions was used to collect data via the Google survey tool [Google Forms]. With informed consent and all inquiries related to the survey in Google Forms, a shareable link was generated. Then, the link was disseminated in different social media platforms (e.g., Facebook, Messenger, WhatsApp, etc.) and to Bangladeshi people whose contact information could be accessed through publicly available telephone directories. The authors also used a face-to-face survey approach to collect data by maintaining social distancing and proper preventive measures to get responses from participants who resided in slum areas or had no access to the internet (*n* = 109). A convenience sampling technique was chosen to recruit the participants, and data were collected from May 2020 to July 2020. Initially, a total of 960 respondents participated in the survey. Finally, 947 valid responses were included in the final analysis (Table 1). The responses were only stored in the Google servers under the authors’ supervision.

For the qualitative data collection, 20 in-depth interviews (IDI) were conducted among the same participants of the quantitative survey. Participants were selected through stratified random sampling techniques (Table 2). These interviews were taken through a telephone survey by the authors (telephone number obtained from the questionnaires). It gave more detailed information than what is available through other data collection methods. Interpretive phenomenological analysis (IPA) was used to analyze the qualitative data. It had an ideographic focus on how the respondents of IDI personally experienced the COVID-19 pandemic in the early lockdown period.

Two types of triangulation methods were used in this study. The first was data source triangulation, which involves the interviewee’s location, communities, and socio-demographic characteristics. Second, between or across method triangulation, i.e., triangulating quantitative (questionnaire survey) and qualitative (IDI transcripts) data. The questionnaire survey and the interviews and secondary data helped to use the triangulation method to validate the survey data [42,43,44].

The inclusion criteria included being Bangladeshi community dwellers and willing to participate in the survey voluntarily, and belonging to the 11–60 age group.

### 2.2. Ethical Approval

The study protocol was reviewed and approved by the Biosafety, Biosecurity, and Ethical review board of the Jahangirnagar University, Savar, Dhaka-1342, Bangladesh [Ref No: BBECJU/M 2021/COVID-19/4(1); dated 26 April 2021]. All participants willingly consented to the survey. Anonymity and confidentiality of data were strictly maintained.

### 2.3. Measures

A self-reported questionnaire including informed consent and questions related to surveys, including socio-demographics and HSBs, was employed during the study.

### 2.4. Socio-Demographic Information

The participants were asked to provide socio-demographic information, including age, gender, education, occupation, monthly income, and residence. All the information was recorded, including several categorical responses (Table 1).

### 2.5. Health-Seeking Behaviors Measure

In total, eight questions were asked to assess the participants’ HSBs concerning consciousness about COVID-19 (yes/no); mode of transport use (public/private/no transport used); mask-wearing (always/occasionally/never); wearing gloves (always/occasionally/never); disinfecting foods before use (always/occasionally/never); ordering food online (yes/no); history of contact with COVID-19 positive cases (yes/no/don’t know); contact with any person that traveled from abroad (yes/no/don’t know); traveled abroad within the last 30 days (yes/no); recent history of attending a public gathering (yes/no); smoking habits (yes/no); and alcohol consumption (yes/no).

### 2.6. Scoring Procedures

A score was generated based on the responses of the study participants. The responses were made dichotomous, and following the seeking, the behavior was scored by “1” (yes) and “0” (no or don’t know) based on the individual seeking behavior. A few questions had three possible responses where priority ranking was required. The total score was obtained by adding the total raw scores of the HSBs. A higher score indicates higher levels of HSBs.

### 2.7. Statistical Analysis

The analysis was performed based on the available data from the self-reported online-based survey, so power calculation was not done. Data were twice checked, cleaned using PASW 22.0 (IBM SPSS Statistics for Windows, Version 22.0. Armonk, NY, USA: IBM Corp.), and analyzed using STATA-15 (StataCorp Limited, College Station, TX, USA); *p*-values were two-sided and considered significant if less than 0.05. COVID-19 awareness related information such as HSB (aware about COVID-19; using a mask; using gloves; mode of transport; disinfected food; food ordered online; H/O contact with active COVID-19 patients; and H/O public gathering) and personal (age and gender) and socio-demographic information (education status, occupation, monthly income, living status, locality, accommodation type) was collected.

Summarized categories were produced for demographic characteristics (age and gender), comorbidity, HSB, and exposure and outcomes. A chi-square test was used to investigate whether significant associations existed between exposure (age, education status, occupation, monthly income, and locality) and outcomes (health-seeking behaviors, HSB). We also investigated the association between the exposures and outcomes using the multivariate regression model.

## 3. Results

### 3.1. General Characteristics of Participants

A total of 947 participants were included in the final analysis. Of them, 60% were male, and the majority ranged from 21 to 30 years (63.5%). The majority (43.8%) had postgraduate or above-average levels of education, whereas 37.8% of participants had an undergraduate level of education. About 30% of the participants in this study were unemployed, and 26.5% were self-employed. Monthly income varied among participants, and 45.7% had <10,000 BDT (USD 117), whereas about 33.6% had >30,000 BDT (351 USD). About 60% resided in common housing areas, 29.5% in modern residential areas, and just over 10% resided in a slum or other areas (Table 1).

### 3.2. Health Seeking Behavior

The distribution of HSBs is presented in Table 3. Most female responders were aware of COVID-19 (94.5%) compared to males (88.2%). About 93.7% of females and 88.1% of males wore masks. Most were aware of the COVID-19 pandemic (90.7%), and a half (49.7%) reported not using any transport during the study period. However, only 4.2% of participants did not wear a mask while going out, whereas nearly half (46.6%) did not use gloves. Among the participants, 68.6% of females and only 48.6% of males disinfected foods before use. The percentages for ordering food from online was 17.7%, contact with COVID-19 positive cases was 10.1%, attendance to a recent public gathering was 30.5%, contact with persons who came from abroad was 3.7%, returned from abroad within the past 30 days was 11.4%, cigarette smoker was 15.8%, and consumed alcohol was 3.6%.

The mean score of HSBs was 9.98 (SD = 2.01) out of 16, with an overall correction rate of 62.4% (Appendix A). HSB in the 11–20 age group (β = −0.76; 95% CI −0.20, −0.96; *p* = 0.010) showed a significantly lower score than those above 50 years of age (Figure 1A, Appendix A). No other age group showed any significant difference when the participants were stratified by gender. Compared to illiterate participants, the participants with higher education levels had higher HSB scores (β = 3.61; 95% CI 2.91, 4.30; *p* < 0.001). S.S.C., H.S.C., undergraduates, and postgraduates or above showed significantly higher HSB knowledge than illiterate participants (Figure 1A, Appendix A). The association remained the same when the participants were stratified by gender, but the magnitude of the association (*p* ≤ 0.002) was higher for female participants (Figure 1B,C, Appendix A). Male participants with an occupation of business (β = −1.61; 95% CI −2.31, −0.90; *p* < 0.001), daily employment (β = −2.85; 95% CI −3.53, −2.17; *p* < 0.001), (Figure 1B), and female daily wager (β = −2.68; 95% CI −4.28, −1.09; *p* < 0.001) and homemaker (β = −1.33; 95% CI −2.06, −0.59; *p* < 0.001) showed significantly lower knowledge about HSB compared to educated unemployed ones. The participants with higher income (>30,000 BDT), male and female, had higher knowledge (β = 0.66; 95% CI 0.46, 0.86; *p* < 0.001) than the participants with an income of <10,000 BDT. When considering the locality, the participants living in modern areas and others had a higher knowledge of health-seeking than the slum-dwelling participants. The association also remained the same after stratified with gender except for other localities for females (Figure 1A–C; Appendix A).

### 3.3. Outcomes of In-Depth Interviews (IDI)

The qualitative results were derived from 20 participants from the quantitative survey, stratified through random sampling representing all the socio-demographic criteria of this study (Table 2). In-depth Interviews (IDI) were taken by telephone from numbers found from the questionnaires and those who agreed for further communication. Some secondary data supporting the observations of the ongoing events were also included in this study. The final results are discussed following based on socio-demographic criteria (age, gender, education, employment, and housing type).

#### 3.3.1. Age-Related Outcomes

Younger respondents (11–30 years) (Table 2, respondents (A, B, C, and E)) took all the preventive measures required but went outside during lockdown for necessary shopping, medical purposes, and jobs. A portion of the young respondents mentioned that they took this responsibility voluntarily because they were worried about the older members of their families, they wanted to protect seniors, and they convinced them to stay home (Table 2, respondents A, C, and D). Most stated that their awareness came through several social media channels, including the international and local printed and electronic news (Table 2, respondents A, B, C, and E).

Older people (>50 years) showed two types of behavior: (1) reluctance to follow social distancing (Table 2, respondents J, L, M, N, and T) guidelines; and (2) following the health safety rules strictly because of their comorbidities and fear of COVID-19 from following the international news (Table 2, respondents G, H, K, and O). They went out for medical and religious purposes but claimed to wear masks and gloves and maintain social distancing (Table 2, respondents Q and R). Therefore, the age factor and its association with HSB are inconsistent and mixed; for example, they wore masks and attended social gatherings. Two people of the same age may not react the same way to the lockdown rules.

#### 3.3.2. Gender-Related Outcomes

This study found that female homemakers are more likely to take precautions (Table 2, respondents G, P, and S). They stated that taking care of their home and family is their primary job. They may not go to the market to shop for food, but they disinfect the foods in an attempt to keep their home and family safe. An exception was seen in using a mask by the homemaker, which was low (data not shown). This is because they did not go out of their house and come into contact with COVID-19 patients because, by the concept of social structure, they are the predominant and primary caretakers of those who are quarantined to the home. Respondent (P) stated, *“I took care of my husband when he got infected. He was isolated, and we used the mask in our home”.*

Most women who have paid jobs bear the double burden of paid and unpaid household work in contrast to working men. So, the “work from home” (WFH) condition has placed an extra burden on females who, in normal times, would have managed in other ways (household help, work-life schedule, etc.). A respondent (B), a corporate banker, said that *“…other time I had a work-home life balance. But now, I have to do WFH while have to prepare food, take care of my children, family, and manage household chores….it is too much, honestly”*. So, when the restaurants offered online food delivery, females ordered food, and as stated in the interviews (Table 2, respondents B and E), it *“…helped reduce the stress of my life”.*

Men attended public gatherings for religious purposes and social responsibilities, such as visiting relatives and attending funerals (Table 2, respondents I, L, M, R, H, and T). In addition, some went outside for recreation purposes such as playing sports, exercising, jogging, traveling, etc. (Table 2, respondents I and M). Nevertheless, respondents who avoided social gatherings still went to funerals to show respect (H). This fact shows that even with access to all relevant information, they cannot avoid social responsibilities or cultural habits.

#### 3.3.3. Education Level Related Outcomes

It was mentioned before that younger people found digital news and print media very useful for COVID-19 preparation; it is the same for educated people. People with no formal or primary education receiving these messages from others failed to realize the overall public health threat of COVID-19 (Table 2, respondents D and J). A lower level of education leads to inadequate knowledge and different attitudes to practices regarding health. One respondent (J) who works as household help in Dhaka city said, *“…all the information I got was from my employer. They told me to wash my hands when I come to work and wear a mask. But I feel uncomfortable (suffocated) when I work while wearing the mask!”* The information by itself was not enough for them. One businessman (T) who operates a local shop said, *“I don’t feel anything will happen to me due to Corona, whereas closing the shop for this situation will definitely affect my life!”.*

Some educated respondents did not follow health safety rules; they chose to take a laidback and casual attitude to the whole situation. One undergraduate respondent (M) said, *“news, social media, various sources are saying various things….it is difficult to understand what should we follow. Some said not to worry about it, and it is just common flu; others are spreading unnecessary fear…They said older people are getting more infected. Does that mean that I am safe?”.*

#### 3.3.4. Employment and Income-Related Outcomes

This study finds that jobs were lost, and businesses were affected by the lockdown in this pandemic. The garment industry, factories, small businesses, local sellers, household helpers, chauffeurs, public transport operators, a rickshaw puller, and day laborers, these people either lost their jobs or had their salaries reduced due to COVID-19 (Table 2, respondents D, J, L, M, N, and T). One respondent (D) stated, *“I had to find alternative solutions to earn and support my family.”* Some migrated to another area to live and find a job even when the lockdown restrictions were active. To quote from the respondent (J), *“my neighbor was 45 years old woman, she worked as household help. But she had to go to her hometown as she lost 3 of her jobs and was unable to pay the rent and afford food.”* Some respondents choose the risk of COVID-19 rather than the threat of losing their job. One respondent (D) said, *“If I can’t work today, my family will die starving rather than COVID!”.*

Some (Table 2, respondents D, J, and L) had to go for relief products and thus were in lines in public for long periods of time. Though lower-income people statistically lack health safety protocols, this study shows that household help, chauffeurs, and part-time workers such as food delivery people, students, shopper’s assistants, etc., follow the COVID-19 precautions as much as possible because their employers told them to do so. In their case, precautions can be masks only or the entire personal protective equipment (PPE) attire, which in most cases was provided by their employers, as seen from our primary data. As their employers’ lifestyle depends on these people, five respondents (Table 2, respondents B, F, H, I, and K) stated that they felt “*socially responsible*” for making their household help, chauffeurs, etc., aware of avoiding the pandemic, thus “*keeping them and their employees safe*”.

#### 3.3.5. Housing Type Related Outcomes

This study found that people from modern and common housing areas (MHA and CHA) take more precautions than those living in slums. It can be explained by the educations, jobs, and incomes of those residing in these areas. One reason can be the administration and authority in these housing settlements. Modern condominiums, apartments, and common housing areas have rules and protocols. A respondent (B) said, *“Our apartment complex has strict mask policy and guest attendance policy. The whole community or neighborhood follows those rules or is bound to follow those rules to live there with harmony and peace. This strict situation may help the people stay safe and stigmatize anyone infected*”. However, there were cases in Dhaka city where infected people and people who worked on the front lines of the pandemic were asked to leave their buildings for the safety of other inhabitants. One of the respondents (K), a physician, stated that *“Initially, neighbors were very scared even to stand next to me or share the elevator with me; which I think is understandable. But my other colleagues have seen worst!”* People who got infected in the early lockdown period said they were scared to tell anyone in fear that their neighbors would shun them (P). So, it is possible that social stigma and rigid protocols can also make people hide information from their neighbors so that they do not get shunned by their neighbors.

This study shows that people living in slums take fewer health safety precautions. Even though some of their jobs require them to wear masks and practice handwashing, they rarely follow these rules in their neighborhood, as stated by responders D and J. Moreover, social isolation is not possible in slum areas. A large family stays in one room, and the environment is very congested. They share a common water source, kitchen, and toilet.

#### 3.3.6. Religion-Related Outcomes

Religious sentiments and ideas significantly impact people’s responses to the lockdown restrictions, as observed in this study. There are observations made from the interviews. Some sought out religion during the pandemic crisis and reacted negatively towards the given rules when asked to stay home. One respondent (T) said, *“Nothing can happen to me in the house of Almighty; he will take care of me!”* Another respondent (M) said, *“If I can go to market for groceries, why can’t I go to say my prayer!*” Some people have very vague knowledge about what to do and not do, from a religious perspective, during the pandemic. As one young respondent (A) stated, *“I felt like I was at war with my father, to make him understand the severity of the situation, and he can say all his prayers at home. Luckily, he understood later on”.* However, on the other hand, some respondents stated that they prayed while isolated in their homes, and their faith gave them hope and clarity to have the patience to endure the pandemic. Respondent (S) said, *“In this time of fear and panic, only my prayers gave me peace”.*

#### 3.3.7. Other Outcomes

Staying in isolation at home gave some of the respondents a claustrophobic feeling, and not being able to leave home to enjoy the outside environment or to be able to see friends and family as a primary source of stress. A respondent (G) stated that *“sometimes I felt difficulty to breathe, thinking about the whole situation and then being stuck in the home…..the anxiety was unimaginable!”* Some reported that they had developed depression due to this lifestyle and issues related to employment. The stress of their family members getting infected was significant. One respondent (F) said, *“Ensuring all family members follow health safety rules is a stress-inducing matter”.*

## 4. Discussion

The present study investigated the relationship between people’s socio-demographic factors (i.e., age, gender, education, occupation, monthly income, and residence) and HSBs (including public adherence to COVID-19 preventive measures and lockdown protocols) in the Bangladeshi community. This paper explored different types of HSB, the socio-demographic factors associated with such behaviors, and the reasons behind any behavioral changes.

### 4.1. Association of Age in Health-Seeking Behavior

During the early stages of the pandemic (March–July 2020), various sources widely circulated that older people were more susceptible to COVID-19 illness and death, and younger people were less at risk [45,46,47]. Knowing the risk, the older participants (>50 years) in this study showed higher scores of HSBs (Figure 1A–C, Appendix A). Whereas the younger generation (21–30, *n* = 603) scored second, and they took all the preventive measures but went outside during the lockdown for essential purposes (Table 2, respondents A, B, C, and E). With reasonably good internet connectivity even in remote areas [48], the young generation of Bangladesh has better access to social media and international and local news than they did even a few years before [48]. The high awareness among the younger community is due to the massive campaigns about COVID-19 vulnerability in the electronic media (Table 2, respondents A, B, C, and E). However, the older people (31–60 years) showed two types of behavior; some followed protective measures (Table 2, respondents G, H, K, and O), a few went outside for medical purposes, some went out for religious reasons with proper precautions (Table 2, respondents Q and R), and some were reluctant or unable to follow the restrictions (Table 2, respondents J, L, M, N, and T). Thus, variable HSB scores were observed along with the age groups (Figure 1).

### 4.2. Association of Gender in Health-Seeking Behavior

There was a stronger association between gender and HSB (Figure 1B,C). In almost every indicator, females had higher scores than males in showing positive HSB. This is in line with other studies [49,50,51,52,53]. A study in Saudi Arabia showed that females showed slightly better practice than males, but males showed more awareness related to COVID-19 [50].

This study showed that homemakers take many precautions (Table 2, respondents G, P, and S), but an exception was seen in wearing a mask by the homemaker, as they did not go out of their house. But they were vulnerable because they were in contact with COVID-19 patients who were being quarantined at home [51]. Nevertheless, the working women responded (Table 2, respondents B and E) that they go through the “double burden” for their responsibilities [54]. That is why working women use private transport and take all the health safety protocols to keep themselves and their family members safe, so that they don’t have to worry about an infected family member and their job and domestic management responsibilities (Table 2, respondents B, E, K, N, and Q) [51].

The study found that males saw taking care of outside work as their responsibility. However, they attended public gatherings for religious purposes and for reasons related to social responsibility, and some went out for recreation purposes as well (Table 2, respondents I, L, M, R, H, and T).

### 4.3. Association of Literacy Level and Access to Information in Health-Seeking Behavior

This study shows that HSB is positively correlated with education levels (Figure 1, Appendix A). Respondents who had an MPhil/Ph.D. level of education were significantly more prepared for COVID-19 than the respondents who had a higher secondary level of education. Similar observations were found by Hossain et al., who concluded that education and individual preparedness build health communication opportunities among the mass population with the most significant importance [52].

People with no formal education take fewer health and safety precautions, and undergraduate or postgraduate people are more sincere about government ensured health safety rules that are disseminated through print and electronic media. This explains that the greater the education, the more access to information and knowledge, as mentioned in the earlier study [55]. Qazi et al. reported that formal education encourages communities to become conscious and aware of the grave threat of COVID-19, thereby enabling them to interpret both national and international news and to be mature enough to differentiate between real and fake news, visual and social media, and to take calculated steps to prepare themselves for this pandemic [56]. However, people without a proper education were unaware of the COVID-19 related threats (Table 2, respondents D and J) [56,57].

Interestingly, even people with higher levels of education sometimes did not follow the COVID-19 restrictions imposed by the government. It shows that despite the knowledge, trust in the policymakers and willingness to follow government rules and regulations is also essential factor that impacts HSB [58]. This mistrust of the respondents regarding rules and regulations imposed by the state can be explained by the mixed opinions about the pandemic from various lines of expertise in the early stage of the pandemic, which made the respondents doubt every piece of information and decision by the experts (Table 2, respondent M).

### 4.4. Association of Financial Solvency and Employment in Health-Seeking Behavior

In the case of income and employment, it is seen from this study that people with medium income tend towards HSB (Figure 1, Appendix A). In this pandemic, some government officials had to attend their office daily or at the scheduled time, wore masks, and used transportation according to the government rules. Many private jobs offered work-from-home opportunities for the safety of their employees. That’s why they went outside for employment less than government jobholders (Table 2, respondents B, E, and I).

This study found that low-income people or daily workers did not wear masks regularly, along with businesspeople (Figure 1; Table 2, respondents’ D and J). They also fall behind in disinfecting outside products (data not shown). Daily workers and some local businessmen did not receive higher education, which explains their lack of information and inability to follow health protocols (Table 3). Also, affording all these health safety products, such as masks, sanitizer, and disinfectants, is difficult for lower-income people, as the study indicates (data not shown). A similar observation was also reported among lower-income and informal jobs [52]. Additionally, during the early stages of the pandemic, the prices of personal protective equipment were very high and scarce across the globe, including in Bangladesh [59].

Many jobs were lost, and businesses were affected by the lockdown in this pandemic situation [59]. The working people or businesspeople either lost their jobs entirely or had reduced incomes due to COVID-19 (Table 2, respondents D, J, L, M, N, and T) [60]. They stated that they had to find alternative solutions to earn and support their family. This statement is in line with other studies conducted in Bangladesh and other LMICs, like Serbia and Egypt [52]. Some migrated to another area to live and find jobs even during the strict lockdown [52,58]. Some had to go for relief products and thus stood in lines in public places for extended periods (Table 2, respondents J, and L). These factors help explain why the lowest health-seeking awareness was observed in lower-income people and businesspeople (Figure 1) [61].

### 4.5. Association of Housing Types in Health-Seeking Behavior

This study found that people from modern and common housing areas take more precautions than those living in slums (Figure 1 and Appendix A). It can be explained by the education, jobs, and incomes of those residing in these areas (Table 3). Social distancing rules vary among people depending on residence area type [62]. One reason could be related to the administration and authority in these housing settlements. Modern condominiums, apartments, and common housing areas have rules and protocols, and the whole community or neighborhood follows those rules or is bound to follow those rules to live there with harmony and peace [52]. However, in Dhaka city, infected people and frontlines were asked to leave their buildings for the safety of other inhabitants [63,64], as reported here by responder K. Therefore, possible social stigma and rigid protocols can also make people hide information from their neighbors so that they do not get shunned or ostracized by them, as stated by responder P. This, therefore, increases the chance of transmission in these types of residences.

On the other hand, this study shows that people living in slums take fewer health and safety precautions (Figure 1). The homeless are at higher risk of COVID-19 because crowded living conditions increase the risk of transmission [64]. The living conditions in slum areas are already crowded, which increases their chance of getting infected [64,65]. This study also found that, with their level of knowledge about COVID-19, their lower incomes, and reduced job rates, people living in slums are less likely to follow COVID-19 restrictions (Table 2, respondents D and J).

### 4.6. Association of Religious Sentiments and Beliefs in Health-Seeking Behavior

Previous studies have shown that optimism and belief about consequences significantly affect behavioral changes [66]. As observed in this study, religious sentiments and ideas significantly affect adherence to lockdowns. Despite strict rules, people reported going to mass religious gatherings to pray at the masjid, temple, and church (Table 2, respondents M and T). Some also had this false idea about social distancing and were unable to adhere to it in spiritual practice (Table 2, respondent M). Some sought out religion in a time of crisis and reacted negatively towards the given rules when asked to stay home (Table 2, respondent T). However, on the other hand, some respondents stated that they prayed within their home isolation, and their faith gave them hope and clarity to have the patience to endure this pandemic (Table 2, respondent S).

### 4.7. Association of Other Factors in Health-Seeking Behavior

People’s obedience towards lockdown protocols can be seen as HSBs [66]. Adherence to these rules depends on people’s choices, behaviors, trust in government and policymakers, and their coordinated decision-making approach [58]. However, some factors affect the implementation of lockdown and are directly related to the socio-demographic aspect of the people, resulting in discouraging people from following health safety rules [66].

In this study, cultural factors were found to have a great deal of influence on HSBs. Bangladesh is a South-Asian country with its own distinctive culture [67]. The culture of Bangladesh is influenced by mainly three religions- Hinduism, Buddhism, and Islam, in successive order, with Islam being the most widely practiced and having the deepest impact [68]. It should be mentioned that Eid-Ul-Fitr, a religious holiday celebrated by Muslims around the globe, is also marked by the majority of Bangladesh, as 90.4% of the population of Bangladesh is Muslim [69]. People who celebrate this festival may be understated, but the respondents still mentioned gatherings (Table 2, respondents I, L, M, Q, R, H, and T). Females went shopping to prepare for the festivals in metropolitan areas like Dhaka and Chattagram (data not shown).

The mental health and strength required to endure a pandemic are very important. In their interviews, respondents stated that one of the difficulties they faced in following health safety rules was their stress and anxiety due to the pandemic (Table 2, respondents’ G and F). A study in Italy [70] showed that half of the subjects felt anxious due to their consumption habits, consumed comfort food, and were persuaded to increase their food intake to feel better. This study found that females were more anxious and disposed to comfort food than males and that they ordered online food more than men to reduce the stresses in their lives (Table 2, respondents B and E).

Tabassum et al. stated that Bangladesh had shown ineffective implementation of strategies and a lack of coordination among ministries at the early stages of the pandemic [58]. People who traveled within the country and came from abroad even after the lockdown was announced demonstrated the lack of coordination between the government authorities. Overcrowded markets became even more so before one of the most prominent Islamic festivals, Eid-ul-Fitr [71]. Despite a public transport shutdown in May, people still went on vacation during the Eid holidays [72].

## 5. Limitations of the Study

The present study possesses several limitations. The causal relationship cannot be explained, as it was a cross-sectional study. Additionally, as the COVID-19 variant changed and its effect had different effects on people from different areas of the society, the outcomes that the study showed from HSB might change over time. Since the data was collected at the early stage of the pandemic, many standards have changed since then. Therefore, if the study had more of an extended timeline to collect data, there may have been a more diversified picture of the reality in Bangladesh. Due to its convenient sampling technique, the findings cannot be regarded as representative of Bangladeshi residents.

Another limitation is that, due to the strict timeframe of the study, only 20 IDIs could be scheduled and conducted. However, these data and available secondary data were used for the quantitative data triangulation. Furthermore, not everyone was accessible via an online survey. Moreover, the average age of the Bangladeshi people is 27 [73], and most responses came from that age group.

Acceptability and accurateness is perceivable variant; they are not horizontal among each group of participants. Most of the indicators are simplified in a common category of explanation. Therefore, the sampling of data and resolution might have a different perspective for the informants.

## 6. Conclusions

The COVID-19 pandemic is a serious challenge for Bangladesh due to resource limitations and its inadequate healthcare system. However, the Bangladesh government has implemented different preventive measures to address this situation, including social distancing, maintaining health etiquette, introducing lockdowns, increasing public awareness, and initiating vaccination campaigns. The findings of this study demonstrate a strong association between an individual’s socio-demographic factors and their level of health safety awareness. People with a lower literacy level cannot follow HSBs and the elderly and educated ones. Therefore, to combat the COVID-19 pandemic successfully in Bangladesh, it is urgent to initiate proper health education and access to information for marginalized and lower-income people.

Moreover, personal protective gear and tests should be financially accessible. An incentive should be provided on a need assessment basis. Finally, the health officials should focus on continuous monitoring and surveillance to obtain high-quality data on socio-demographic factors to implement the public health measures.

## Figures and Tables

**Figure 1 healthcare-10-00483-f001:**
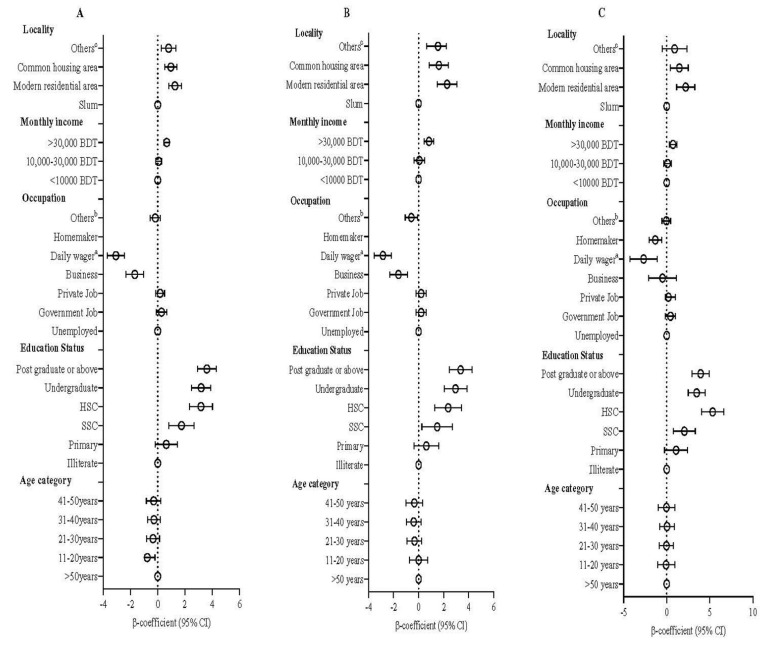
Association of socio-demographic factors and predicting HSB during the pandemic in the whole study population (**A**), Male (**B**), and Female (**C**). The multivariate regression model was used to estimate the *p*-value. BDT = Bangladeshi Taka (Equivalent to 0.012 USD), ^a^ Rickshaw puller, Day laborer, etc.; ^b^ House tutor, part-time, etc.; ^c^ Village, hostel, dormitory, etc.

**Table 1 healthcare-10-00483-t001:** Socio-demographic profile of the study participants.

Variables	*n*	(%)
Age		
11–20 years	52	(5.5)
21–30 years	603	(63.7)
31–40 years	198	(20.9)
41–50 years	60	(6.3)
>50 years	34	(3.6)
Sex		
Male	568	(60.0)
Female	379	(40.0)
Education		
Illiterate	27	(2.9)
Primary	67	(7.1)
SSC	31	(3.3)
HSC	49	(5.2)
Undergraduate	359	(37.8)
Post graduate or above	414	(43.8)
Occupation		
Government Job	131	(13.8)
Private Job	251	(26.5)
Business	37	(3.9)
Daily wager ^a^	40	(4.2)
Home maker	28	(3.0)
Unemployed	281	(29.7)
Others ^b^	179	(18.9)
Monthly income		
<10,000 BDT	366	(45.7)
10,000–30,000 BDT	166	(20.7)
>30,000 BDT	269	(33.6)
Locality		
Slum	39	(4.1)
Modern residential area	280	(29.5)
Common housing area	572	(60.2)
Others ^c^	59	(6.2)

Note: BDT = Bangladeshi Taka; ^a^ Rickshaw puller, Day laborer, etc.; ^b^ House tutor, part-time, etc.; ^c^ Village, hostel, dormitory, etc.

**Table 2 healthcare-10-00483-t002:** Participant list for in-depth-interview selected based on socio-demographic characteristics from the online and face-to-face survey participants.

Respondent Identification No.	Age (Years)	Gender	Education	Job Type	Income (BDT)	House Type
A	11–20	M	HSC	Student	Dependent	CHA
B	21–30	F	Graduate	Private	>30,000	MHA
C	21–30	F	Undergraduate	Govt. Job	10,000–30,000	CHA
D	21–30	M	Illiterate	Daily wager	<10,000	Slum
E	21–30	F	Graduate	Private	10,000–30,000	CHA
F	31–40	M	Graduate	Private	>30,000	CHA
G	31–40	F	Undergraduate	Home maker	Dependent	CHA
H	31–40	M	>Post graduate	Govt. Job	>30,000	MHA
I	31–40	M	>Post graduate	Private	>30,000	CHA
J	31–40	F	Illiterate	Daily wager	<10,000	Slum
K	31–40	F	>Post graduate	Private	>30,000	MHA
L	31–40	M	HSC	Private	10,000–30,000	CHA
M	41–50	M	Undergraduate	Business	>30,000	CHA
N	41–50	F	Graduate	Business	>30,000	CHA
O	41–50	M	Graduate	Govt. Job	>30,000	CHA
P	51–60	F	Undergraduate	Home maker	Dependent	CHA
Q	51–60	F	Graduate	Private	>30,000	MHA
R	51–60	M	Undergraduate	Retired	>30,000	MHA
S	51–60	F	Graduate	Home Maker	Dependent	CHA
T	51–60	M	HSC	Business	>30,000	CHA

Note-M = Male; F = Female; Govt. = Government; CHA = Common housing area; MHA = Modern Housing Area.

**Table 3 healthcare-10-00483-t003:** Health-seeking behavior of the study participants.

Variables	Overall *n* (%)	Male *n* (%)	Female *n* (%)
Aware about COVID-19			
Yes (1)	859 (90.7)	501 (88.2)	358 (94.5)
No (0)	88 (9.30)	67 (11.8)	21 (5.50)
Mode of transport use			
Public (0)	221 (23.3)	146 (25.6)	75 (19.7)
Private (1)	255 (26.9)	166 (29.2)	89 (23.5)
No transport use (2)	471 (49.7)	256 (45.1)	215 (56.7)
Wear mask			
Always (2)	845 (89.2)	490 (86.3)	355 (93.7)
Occasionally (1)	62 (6.50)	50 (8.8)	12 (3.3)
Never (0)	40 (4.20%)	28 (4.90)	12 (3.2)
Use of hand gloves			
Yes (2)	293 (30.9)	134 (23.6)	159 (42.0)
No (0)	441 (46.6)	305 (53.7)	136 (35.9)
Occasionally (1)	213 (22.5)	129 (22.7)	84 (22.2)
Disinfected foods before use			
Always (2)	536 (56.6)	276 (48.6)	260 (68.6)
Occasionally (1)	194 (20.5)	137 (24.0)	57 (15.0)
Never (0)	217 (22.9)	155 (27.3)	62 (16.4)
Ordered food online			
Yes (1)	168 (17.7)	79 (13.9)	89 (23.5)
No (0)	779 (82.3)	489 (86.1)	290 (76.5)
Join public gathering			
Yes (0)	289 (30.5)	214 (37.5)	75 (19.7)
No (1)	658 (69.5)	354 (62.3)	304 (80.2)
Contact with active cases			
Yes (0)	96 (10.1)	58 (10.2)	38 (10.0)
No (1)	500 (52.8)	308 (54.2)	192 (50.7)
Don’t know	351 (37.1)	202 (35.6)	149 (39.3)
Contact with persons who came from Abroad within 30 days			
Yes (0)	108 (11.4%)	82 (14.4%)	26 (6.90%)
No (1)	839 (88.6%)	486 (85.6%)	353 (93.1%)
Travel abroad within 30 days			
Yes (0)	35 (3.70)	20 (3.5)	15 (4.0)
No (1)	912 (96.3)	548 (96.5)	363 (96.0)
History of smoking			
Yes (0)	150 (15.8)	143 (25.2)	7 (1.8)
No (1)	797 (84.2)	425 (74.8)	372 (98.2)
History of consuming alcohol			
Yes (0)	34 (3.6)	29 (5.1)	5 (1.3)
No (1)	913 (96.4)	539 (94.9)	374 (98.7)

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
