# Peer review of "The Effect of Socio-Demographic Factors in Health-Seeking Behaviors among Bangladeshi Residents during the First Wave of COVID-19"

_healthcare, 2022, doi:10.3390/healthcare10030483_

Round 1

Reviewer 1 Report

See attachment

Author Response

Reviewer 1: Comments and Authors Response

Comment-1 The abstract is not exactly the summary of the study and need some adaption. You write about online cross-sectional study but in the methods, you also listened face-to-face data sampling methodology.

Response: We would like to thank the reviewer for the critical observations. We have re-written the whole abstract based on the findings of the study (lines 37-58). We have performed a cross-sectional study with a mixed methodology. Convenience sampling techniques were used for online surveys, purposive sampling technique was used for a face-to-face interview, and stratified random sampling technique was used for an in-depth interview.

Comment-2 In the introduction it would be very helpful to describe the Bangladesh society and add some demographical data about the society. I miss also the % of fully vaccinated people in line 99. In line 105-106 is not clear what would you like to say. Line 109 has different references as later mentioned at describing the steps, so please add the steps you write about. In line 133 you mentioned coping strategies – which have you thought about?

Response: We would like to thank the reviewer for the suggestions. We have added a short description of Bangladesh in the introduction section (lines- 70-75). Moreover, according to the reviewer's recommendations in the introduction section, we have included Bangladesh's current COVID-19 vaccination status (lines 99-101). We have re-written lines 105-106, as "Public awareness is the prerequisite to implementing the control interventions for combatting the COVID-19 pandemic; however, such awareness is governed by an individual's health safety practices and etiquettes", (new lines 107-109). We have re-adjusted the position of the references of line 109 and described the steps briefly, as recommended by the reviewer (lines 113-115). A coping strategy could be the initiation of health education, especially for the poor, uneducated people, to maintain health precautions such as wearing masks, to reduce the transmission of COVID-19.

Comment-3 Methods: this section is a bit confused, you mentioned text messages (LINE 155-156) but is not clear how could you use this possibility to introduce the research. Data protection and availability of the phone numbers of the participants are not described. I also miss some description of the participants for the qualitative part of the research. Some parts are repeat more then once (line 201-207).

Response: We would like to thank the reviewer for the careful observations. The online questionnaire was disseminated through various social networking platforms. Moreover, universities/offices/healthcare facilities were contacted and requested through telephone to share the questionnaire among the stakeholders (the numbers were obtained from the public directory). The authors only accessed the responses, and only the authors performed the face-to-face interviews to maintain data confidentiality. The method section is edited accordingly (lines 154-167). We have also elaborated the qualitative method section for more clarification of data sampling, storage, and confidentiality (lines 173-179). Lines 201-207 (new lines 220-226) explain which data were used as exposure and outcomes for analysis purposes. The section seems to be repeated but carries important information about the data analysis.

Comment-4 Results: You didn't add anything about the qualitative part, I miss the description of the participants. The average participants are quite young. Is this in corelation with the population in Bangladesh or the region where the research was conducted? Could you please add the main outcomes (theme, categories, code) from the qualitative part of your study or please withdraw that part from the article.

Response: We are grateful to the reviewer for such a wonderful observation and suggestions. We are not sure how we missed this part. We have added the qualitative part in the result section (lines 283-442). The average age of Bangladesh is 27 years (https://www.worldometers.info/world-population/bangladesh-population/). Moreover, the young generation is more prone to use the internet than other ages. As a result, the questionnaire was more accessible for this group. We have mentioned this limitation section (lines). We thank the reviewer for the recommendations. We have categorically described the outcomes of the survey in the result section (lines 283-442).

Comment-5 - Line 296-298 is not clear what you would like to say Response: We have tried to explain the issue in lines 321-327 and 456-458 with reference. - Line 339 please add a reference – it is a strong statement.

Response: We have re-written the lines 483-485 - Line 412, are the tests free?

Response: We have re-written the paragraph, and the line was omitted. - Line 439, please explain – what means "own distinctive culture" Response: We have discussed the section in the edited manuscript (lines 559-562). - Line 458, are the data from qualitative or quantitative source?

Response: The observations were from IDIs and discussed in lines 540-551with quoting the respondents. - Line 472, please add some typically statement from the participants of the qualitative part of research

Response: We would like to thank the reviewer for the suggestions. We have added a few statements in the result sections (lines 319-320, 325-329, 343-348, 350-354, 360-366, 376, 383-386, 389-390, 404-413, and 417-422). - Line 486, do you have any reference of that statement?

Response: We have found the observation from the qualitative data and added the reference (lines 570-576) - Line 501, that you have already mentioned Response: The lines have been removed from the manuscript. The Limitation of the study should be partly depended on the introduction of the population where the research was done.

Response: We have edited the limitation section according to our observations (Lines 584-607) The conclusion should be stronger related on your main findings. Response: We have edited the limitation section according to our observations (Lines 610-623). 

Reviewer 2 Report

This is a study on the determinants of help-seeking behavior among citizens of Bangladesh by using a convenience sample.  The Introduction looks all right but after that guidelines for scientific reporting in international journals are not followed. The language needs extensive editing after the Introduction throughout the paper with missing words making the sentence incomprehensible for an external reader. As an example I cite the description of the scoring procedure of the main outcome variable, i.e. help-seeking behavior 'A score was generated based on the response of study participants. The response was dichotomous and followed the seeking behavior was scored “1 and “0” if not or don’t know. A few questions with 3 options to respond to, based on priority 2, 1, and 0 scores was noted'.

Based on this descripition I cannot conclude how the scoring was carried out.

Moreover, the results are given in a very condensed and brief format but results with various variables that are not mentioned in the Results chapter are later discussed under specific sub-headings. The results are mainly expected and self-evident. The discussion of them is rather trivial and leaves the potential bias of the convenience sample unmentioned.  

Author Response

Reviewer 2: Comments and Authors Response

This is a study on the determinants of help-seeking behavior among citizens of Bangladesh by using a convenience sample.  The Introduction looks all right but after that guidelines for scientific reporting in international journals are not followed. The language needs extensive editing after the Introduction throughout the paper with missing words making the sentence incomprehensible for an external reader. As an example I cite the description of the scoring procedure of the main outcome variable, i.e. help-seeking behavior 'A score was generated based on the response of study participants. The response was dichotomous and followed the seeking behavior was scored "1 and "0" if not or don't know. A few questions with 3 options to respond to, based on priority 2, 1, and 0 scores was noted'.

Based on this description I cannot conclude how the scoring was carried out.

Response: We would like to thank the reviewer for his careful reading and suggestions. We have edited the whole manuscript. The scoring section has been edited (lines 210-215).

Moreover, the results are given in a very condensed and brief format, but results with various variables that are not mentioned in the Results chapter are later discussed under specific sub-headings. The results are mainly expected and self-evident. The discussion of them is rather trivial and leaves the potential bias of the convenience sample unmentioned.

Response: We think the reviewer has correctly pointed out the missing part of the manuscript. We have elaborately described the observations from the qualitative study and discussed them based on our quantitative and qualitative observations only (lines 283-583).

Reviewer 3 Report

Thanks for the interesting paper. My comments can be seen in the attachment.

Author Response

Reviewer 3: Comments and Response

Comment 1:

not relevant to the subject of the paper

Response: We would like to thank the reviewer for carefully checking the manuscript. The sentence has been omitted from the manuscript, and new sentences have been added to describe Bangladesh society (lines 70-75).

Comment 2:

How many received a first course? A second course?

The number of vaccinated is out of how many potential candidates? Are the vaccines free and accessible to all populations?

Response: The vaccination program is free of cost and available to all people. As of February 3, 2022, 57.74% and 37.12% of the Bangladeshi people have received their first dose of the Covid-19 vaccine, respectively, whereas only 2.82% of the vast population have been administered with the third dose. The information is added in the manuscript (lines 99-101).

Comment 3:

How were they recruited?

Response: The study was conducted during the early stage of the pandemic, and during the survey, it was noted that one class of people was missing, i.e., illiterate, daily wagers, etc. Thus, to collect some data from illiterate, daily wagers, and floating people, the authors personally collected the data from the streets with proper precautions (lines 161-164).

Comment 4:

How were they identified, and on what basis were they selected? The quantitative survey was not anonymous?

Response: The responders were selected using stratified random sampling techniques. They were selected from the online survey responders or face-to-face interviews to assess all socio-demographic classes. The responders' description is discussed in the method section and Table-2 (lines 173-183).

Comment 5:

Comment: Who interviewed them?

Response: The authors conducted the interviews.

Comment 6:

Where was the questionnaire taken from? Was it written by the authors? How is the questionnaire valid?

Response: We would like to thank the reviewer for the comment. The authors developed the questionnaire for the survey. Before circulating the questionnaire, the authors checked face validity, conducted a pilot test, and checked the question's internal consistency. Later authors revised the questions based on the validity checking outcomes. The authors disseminated the questionnaire online to the university student, office staff, and health workers during the validity checking. The authors also regularly checked the COVID-19 health-seeking guideline of CDC and WHO.

Comment 7:

What is the range of the new variable? Zero up ..?

Response: The scoring method is explained in lines 210-215 and Table -3.

Comment 7:

Comment: It seems that these are the findings of a quantitative study. What about in-depth interviews?

Response: We would like to thank the reviewers for the critical observation. We have missed this part in our manuscript and are grateful to the reviewer for addressing it. We have added a qualitative result section in our manuscript (lines 283-442).

Comment 8:

Comment: What is the mean age and SD?

Response: We collected age as a categorical variable thus, SD will not be estimated.

Comment 9:

Is it above or below average?

Response: We thanked the reviewer for pointing it out. Yes, this is below the average, and it is only USD 117. Now we have been added this in the main text.

Comment 10:

Comment: I did not find the table

Response: We have prepared the tables and figures during the submission, but unfortunately, those were missed from the manuscript during formatting to journal style. We have added Tables-1,2 and Figure-1 in the manuscript. Also, we have added supplementary tables-1 and 2.   

Round 2

Reviewer 1 Report

The authors improved the article and so it can be published at the Journal.

Author Response

Dear Dr. Ashlyn Shi
Assistant Editor  

Healthcare

We want to thank you for your consideration of our manuscript 'The Effect of Socio-demographic Factors in Health-Seeking Behaviors Among Bangladeshi Residents During the First Wave of COVID-19' and for the feedback provided by the reviewer. We have now revised the manuscript addressing the reviewer's comments. Please find below our point-by-point response to the critiques. We hope the paper will now be acceptable for publication in Healthcare.

Responses to Reviewer 1

Reviewer's comment: The authors improved the article to be published in the Journal.

Response: We would like to thank the reviewer for the positive comments.

Responses to Reviewer 2

Reviewer's comment: The submission has much improved compared to its earlier version. However, it still fails to meet the criteria for acceptance in an international referee-based scientific Journal. Here are some main points. The in-depth interviews are mentioned in the Abstract but nothing of their results. It is hard for an external reader to conclude whether the results reported in the Abstract originate from the in-depth interviews or the logistic regression analysis but most probably in the later ones. My major criticism, however, is directed towards the analysis of the interview data as it is focused on reporting the same statistical association already reported in the quantitative analysis, which is not at all the point of the triangulation methodology that should aid to identify new dimensions of the results that cannot be sufficiently described utilizing quantitative statistical analysis. For this purpose, some qualitative analysis methodology should have been applied, e.g., content analysis. No such theoretical framework is referred to in the manuscript.

Response:

We appreciate the reviewer's comments and suggestions. We have added qualitative findings in the Abstract (Lines 56-60).

"The study was conducted during the lockdown state of the first wave of Bangladesh in 2020. It was not possible to perform a simple random sampling techniques-based study. Thus, online survey-based convenience sampling was performed. However, IDI was not focused on the same statistical data. Rather, it was developed to get in-depth data that could not be derived through the online survey. Though there were some open-ended questions in the online survey, very few responded to those, which led to the IDI and the authors took the interviews based on a checklist, which mainly emphasized people's experience during the lockdown, their behavior, and their "whys" and "hows" of those activities during COVID-19 lockdown. Thus, our qualitative data does not always match the quantitative data. A few observations were solely collected from qualitative data and described in the result and the discussion section.

For example, in the result section, lines 307-323, we have observed that, though quantitative data shows older people (>50 years) were more aware, the younger generation took the responsibilities of older people and went out. However, many older people reported going out while maintaining safety precautions. Thus, qualitative data found opposite observations to the quantitative ones.

Again, in gender-based observations, although the females were scored higher in all aspects of HSB, they took the responsibilities of the family members and took care of the COVID-19 infected individuals of the family (observation came from qualitative data, line 328-333). Moreover, females were stressed due to "Double-burden" and ordered online food, an observation found in quantitative data, but the reasons found in qualitative data (334-342).

In the case of education-level-based observation, although higher education correlated with higher HSB, a few respondents with higher education levels found not to follow safety rules because of losing jobs, maintaining family, or not trusting the COVID-19 news (358-366).

Though lower-income people in this study scored less in HSB, a few reasons behind those were found from the qualitative research (368-378). A similar type of observation is observed for other factors also.

Thus, the qualitative data is not always correlated with the quantitative observations and is extensively discussed in the discussion section. Moreover, in the method section, the data triangulation protocol used in this study is discussed (Lines 183-191).

Interpretive Phenomenological Analysis (IPA) was used to analyze the qualitative data. It had an ideographic focus on how the respondents of IDI personally experienced the covid-19 pandemic and early lockdown period.

Two types of triangulation methods were used in this study. First, data source triangulation, that is, location, communities, socio-demographic characteristics of the interviewee. And second, between or across method triangulation, i.e., triangulating quantitative (questionnaire survey) and qualitative (IDI transcripts) data. The questionnaire survey, the interviews, and the secondary data helped to use the triangulation method to validate the survey data.

Reviewer's comment: The major weakness of the Discussion is that it does not deal with the cross-sectional design of the study, which again can give wrong interpretations of the associations found. As an example, I could refer to the subheading 'Influence of Age in Health-Seeking Behavior' in the Discussion as the relation theoretically can also be the other way round as old people showed better compliance to good health behavior and hence, have greater chances of survival, particularly in times of a pandemic. In the Discussion, the authors mention the convenience sample as a potential limitation but fail to speculate on how this might have influenced the present results and, via that, the conclusions and the generalizability of the study results.

Response: Thanks for your detailed comments. We have now added more limitations in the Discussion as to the reviewer's suggestions-

"The present study had several limitations which should be acknowledged. The causal relationship can not be explained as it was a cross-sectional study. A prospective study would overcome this limitation."(lines 596-598)

"Due to its' convenience sampling technique, the findings can not be regarded as representative to Bangladeshi residents. A simple random sampling technique would be more appropriate in this regard."(lines 603-605)

Reviewer's comment: The manuscript's language has improved, but I still detected several linguistic errors throughout the paper.

Response: We have now proofread the manuscript.

Responses to Reviewer 3

Reviewer's comment: Thank you for the revised version. All the vague points have been explained in detail, and now the paper is written much better and clearer.

Response:

We want to thank the reviewer for the suggestions to improve the quality of the manuscript.  

Reviewer 2 Report

The submission has much improved in comparison to its earlier version. However, it still fails to meet the criteria for acceptance in an international referee based scientific journal. Here are some main points. The in-depth interviews are mentioned in the Abstract but nothing of their results. In fact, it is hard for an external reader to conclude whether the results reported in the Abstract originate from the in-depth interviews or the logistic regression analysis but most probably in the latter ones. My major criticism, however, is directed towards the analysis of the interview data as it is focussed on reporting the same statistical association already reported in the quantitative analysis which is not at all the point of the triangulation methodology who should aid to indentify new dimensions of the results that cannot be sufficiently described by means of a statistical quantitative analysis. For this purpose some qualitative analysis methodology should have been applied as e.g. content analysis. No such theoretical framework is referred to in the manuscript.

The major weakness of the Discussion is that it does not deal with the cross-sectional design of the study which again can give wrong interpretations of the associations found. As an example I could refer to the subheading 'Influence of Age in Health-Seeking Behavior' in the Discussion as the relation theoretically can also be the other way round as old people showed better compliance to good health behavior and hence, have geater chances of survival, particularly in times of a pandemic. In the Discussion the authors mention the convenience sample as a potential limtation but fail to speculate in how this might have influenced the present results and via that the conclusions and the generalizability of the study results.

The language of the manuscript has improved but I still detected several linguistic errors throughout the paper. 

I am sorry for not being able to give a more positive evaluation of the manuscript but it is the duty of the reviewer, to give a neutral and scientifically sound but critical evaluation of a submission.

Author Response

Responses to Reviewer 2

Reviewer's comment: The submission has much improved compared to its earlier version. However, it still fails to meet the criteria for acceptance in an international referee-based scientific Journal. Here are some main points. The in-depth interviews are mentioned in the Abstract but nothing of their results. It is hard for an external reader to conclude whether the results reported in the Abstract originate from the in-depth interviews or the logistic regression analysis but most probably in the later ones. My major criticism, however, is directed towards the analysis of the interview data as it is focused on reporting the same statistical association already reported in the quantitative analysis, which is not at all the point of the triangulation methodology that should aid to identify new dimensions of the results that cannot be sufficiently described utilizing quantitative statistical analysis. For this purpose, some qualitative analysis methodology should have been applied, e.g., content analysis. No such theoretical framework is referred to in the manuscript.

Response:

We appreciate the reviewer's comments and suggestions. We have added qualitative findings in the Abstract (Lines 56-60).

"The study was conducted during the lockdown state of the first wave of Bangladesh in 2020. It was not possible to perform a simple random sampling techniques-based study. Thus, online survey-based convenience sampling was performed. However, IDI was not focused on the same statistical data. Rather, it was developed to get in-depth data that could not be derived through the online survey. Though there were some open-ended questions in the online survey, very few responded to those, which led to the IDI and the authors took the interviews based on a checklist, which mainly emphasized people's experience during the lockdown, their behavior, and their "whys" and "hows" of those activities during COVID-19 lockdown. Thus, our qualitative data does not always match the quantitative data. A few observations were solely collected from qualitative data and described in the result and the discussion section.

For example, in the result section, lines 307-323, we have observed that, though quantitative data shows older people (>50 years) were more aware, the younger generation took the responsibilities of older people and went out. However, many older people reported going out while maintaining safety precautions. Thus, qualitative data found opposite observations to the quantitative ones.

Again, in gender-based observations, although the females were scored higher in all aspects of HSB, they took the responsibilities of the family members and took care of the COVID-19 infected individuals of the family (observation came from qualitative data, line 328-333). Moreover, females were stressed due to "Double-burden" and ordered online food, an observation found in quantitative data, but the reasons found in qualitative data (334-342).

In the case of education-level-based observation, although higher education correlated with higher HSB, a few respondents with higher education levels found not to follow safety rules because of losing jobs, maintaining family, or not trusting the COVID-19 news (358-366).

Though lower-income people in this study scored less in HSB, a few reasons behind those were found from the qualitative research (368-378). A similar type of observation is observed for other factors also.

Thus, the qualitative data is not always correlated with the quantitative observations and is extensively discussed in the discussion section. Moreover, in the method section, the data triangulation protocol used in this study is discussed (Lines 183-191).

Interpretive Phenomenological Analysis (IPA) was used to analyze the qualitative data. It had an ideographic focus on how the respondents of IDI personally experienced the covid-19 pandemic and early lockdown period.

Two types of triangulation methods were used in this study. First, data source triangulation, that is, location, communities, socio-demographic characteristics of the interviewee. And second, between or across method triangulation, i.e., triangulating quantitative (questionnaire survey) and qualitative (IDI transcripts) data. The questionnaire survey, the interviews, and the secondary data helped to use the triangulation method to validate the survey data.

Reviewer's comment: The major weakness of the Discussion is that it does not deal with the cross-sectional design of the study, which again can give wrong interpretations of the associations found. As an example, I could refer to the subheading 'Influence of Age in Health-Seeking Behavior' in the Discussion as the relation theoretically can also be the other way round as old people showed better compliance to good health behavior and hence, have greater chances of survival, particularly in times of a pandemic. In the Discussion, the authors mention the convenience sample as a potential limitation but fail to speculate on how this might have influenced the present results and, via that, the conclusions and the generalizability of the study results.

Response: Thanks for your detailed comments. We have now added more limitations in the Discussion as to the reviewer's suggestions-

"The present study had several limitations which should be acknowledged. The causal relationship can not be explained as it was a cross-sectional study. A prospective study would overcome this limitation."(lines 596-598)

"Due to its' convenience sampling technique, the findings can not be regarded as representative to Bangladeshi residents. A simple random sampling technique would be more appropriate in this regard."(lines 603-605)

Reviewer's comment: The manuscript's language has improved, but I still detected several linguistic errors throughout the paper.

Response: We have now proofread the manuscript.

Reviewer 3 Report

Thank you for the revised version. All the vague points have been explained in detail and now the paper is written much better and clearer.

Author Response

(The authors gave the same response as above.)

Round 3

Reviewer 2 Report

The manuscript has again improved. However, I still have two comments that should be taken into consideration before the manuscript from my perspective could be accepted for publication.

First, give at least one scientific reference for the triangulation method, preferably more than one.

Revise the sub-headings in the Discussion from containing the term 'Influence' and replace it with 'Association' due to the cross-sectional design of the study.

Author Response

Reviewer II

Round III

Open Review

English language and style

() Extensive editing of English language and style required
() Moderate English changes required
(x) English language and style are fine/minor spell check required
( ) I don't feel qualified to judge about the English language and style

Yes

Can be improved

Must be improved

Not applicable

Does the introduction provide sufficient background and include all relevant references?

(x)

( )

( )

( )

Is the research design appropriate?

( )

(x)

( )

( )

Are the methods adequately described?

( )

(x)

( )

( )

Are the results clearly presented?

( )

(x)

( )

( )

Are the conclusions supported by the results?

(x)

( )

( )

( )

Comments and Suggestions for Authors

The manuscript has again improved. However, I still have two comments that should be taken into consideration before the manuscript from my perspective could be accepted for publication.

First, give at least one scientific reference for the triangulation method, preferably more than one.

Thanks Sir. We have added THREE references.

Reference

  1. Carter, N.; Bryant-Lukosius, D.; DiCenso, A.; Blythe, J.; Neville, A.J. The use of triangulation in qualitative research. Oncol Nurs Forum 2014,41(5),545-7.
  2. Briller, S.H.; Meert, K.L.; Schim, S.M.; Thurston, C.S.; Kabel, A. Implementing a triangulation protocol in bereavement research: a methodological discussion. Omega (Westport) 2008,57(3),245-60.
  3. Bekhet, A. K.; Zauszniewski, J. A. Methodological triangulation: An approach to understanding data. Nurse Res 2012 20(2), 40-43.

Revise the sub-headings in the Discussion from containing the term 'Influence' and replace it with 'Association' due to the cross-sectional design of the study.

Thanks. We replaced as per advice.

Submission Date

17 January 2022

Date of this review

24 Feb 2022 09:59:23